# Extending the Validity of Squeeze Film Damping Models with Lower Aspect Ratios

**DOI:** 10.3390/s22031054

**Published:** 2022-01-29

**Authors:** Xiang Xu, Weidong Fang, Jian Bai, Jiaxiao Chen, Yuan Yao, Qianbo Lu

**Affiliations:** 1State Key Laboratory of Modern Optical Instrumentation, Zhejiang University, Hangzhou 310027, China; iconyang@zju.edu.cn (X.X.); fangwd@zju.edu.cn (W.F.); jiaxiaochen@zju.edu.cn (J.C.); 2Huazhong Institute of Electro-Optics-Wuhan National Lab. for Optoelectronics, Wuhan 430074, China; opt_yaoyuan@zju.edu.cn; 3Frontiers Science Center for Flexible Electronics (FSCFE), MIIT Key Laboratory of Flexible Electronics (KLoFE), Shaanxi Key Laboratory of Flexible Electronics (KLoFE), Institute of Flexible Electronics (IFE), Ningbo Institute of Northwestern Polytechnical University, Northwestern Polytechnical University, Xi’an 710072, China; iamqlu@nwpu.edu.cn

**Keywords:** MEMS, squeeze film air damping model, low aspect ratio, Reynolds equation, finite size effect

## Abstract

Squeeze film air damping is a significant factor in the design of MEMS devices owing to its great impact on the dynamic performance of vibrating structures. However, the traditional theoretical results of squeeze film air damping are derived from the Reynolds equation, wherein there exists a deviation from the true results, especially in low aspect ratios. While expensive efforts have been undertaken to prove that this deviation is caused by the neglect of pressure change across the film, a quantitative study has remained elusive. This paper focuses on the investigation of the finite size effect of squeeze film air damping and conducts numerical research using a set of simulations. A modified expression is extended to lower aspect ratio conditions from the original model of squeeze film air damping. The new quick-calculating formulas based on the simulation results reproduce the squeeze film air damping with a finite size effect accurately with a maximum error of less than 1% in the model without a border effect and 10.185% in the compact model with a border effect. The high consistency between the new formulas and simulation results shows that the finite size effect was adequately considered, which offers a previously unattainable precise damping design guide for MEMS devices.

## 1. Introduction

The squeeze film air damping effect occurs when a plate is pushed towards a rigid surface with a fluid film in between. Squeeze film air damping has a strong influence on the dynamic behavior of non-vacuum microelectromechanical devices, such as the quality factor of micro-resonators [1,2], contacting time of micro-switchs [3], bandwidth of MEMS accelerometers [4,5], and frequency response of electric bearings [6,7]. An extensive study of the models for squeeze film air damping has been performed in past years, both analytically and numerically [8,9,10,11], which are mainly based on the Reynolds equation.

The Reynolds equation was introduced by Tipei half a century ago [12]. It is a good approximation of Navier–Stokes equations under the conditions of a small Reynolds number and sufficiently large ratios of structure dimension to fluid film thickness in general cases. However, this simplification tends to bring notable errors for the microsystem components [13,14], whose ratio of the plate width to film thickness is small. In such cases, the damping effect derived from the Reynolds equation tends to be underestimated owing to the neglect of the border effect and finite size effect.

Expensive efforts have been undertaken to investigate the border effect in past decades [15,16,17], while less attention has been paid to the finite size effect caused by the pressure change across the film. Langlois [18] firstly mentioned the finite size effect in 1962, stating that the pressure across the film may vary significantly when the length of the plate is not larger enough than the film thickness, which violates the assumption of the general form of the Reynolds equation. Satish Vemuri [19] developed a low-order behavioral squeeze film model incorporating both the border effect and the finite size effect in 2000, but this model fails to give the influence of the finite size effect alone, and the ratio of plate length to film thickness is at least 2.5, with applied dimensions only in the range of a few microns. Gabriele Schrag [20,21] proposed a mixed level system simulation for squeeze film air damping realized in VHDL-AMS, a language supporting systematic simulation, in 2001. With error compensation for the finite size effect, the overall accuracy of the model was improved. However, these previous works have not given a quantitative calculation model of damping coefficient considering the finite size effect, which is very important in the dynamic performance design of MEMS devices.

In this paper, we extend the validity of the squeeze film air damping model to a lower aspect ratio by combining the original theory model with the finite size effect. We first introduce the basic theory by utilizing the Navier–Stokes equation and Reynolds equation in squeeze film air damping, and discuss the deviation of the finite size effect in simplification. Thereafter, a series of simulation models are built, including a simple model with the finite size effect only and a compact model with the finite size effect and border effect. Then, we further develop compact quick-calculating formulas of damping coefficient based on the simulation results and general solutions of the Reynolds equation. As the new formulas are based on simulations of scalable parameters, they will be a fast and strong guide in the damping-related analysis and corresponding MEMS design.

## 2. Theory of Squeeze Film Air Damping

The Navier–Stokes equation, which was introduced by Navier in 1821 and Stokes in 1845 [22], has been widely used in the dynamics of viscous flow. The most general form of the Navier–Stokes equation is
(1)$$\rho \frac{DV}{Dt}=\rho f-\nabla p+\mu \nabla^{2}V,$$
where *ρ* is the fluid density, *p* is the pressure, *V* is the fluid velocity, *μ* is the viscous coefficient, and *f* the external force.

The air between the surfaces can be described as an incompressible flow in most conditions [23], when the relative motion of the surfaces squeezes a fluid film between two parallel surfaces with a small squeeze number *σ* (which is defined as $\sigma =\frac{12\mu \omega l^{2}}{P_{a}h_{0}^{2}}$, where *l* is the characteristic length of the plate, *h*_0_ is the initial film thickness, *μ* is the viscosity coefficient of the fluid, *P_a_* the ambient pressure, and *ω* is the radial frequency). Within the range of interest of gas lubrication theory, the viscosity coefficients *μ* and *λ* can be assumed to be constant. Thus, when a plate with length *l* moves towards a stationary wall and the initial fluid thickness is *h*_0_, the Navier–Stokes equation governing the behavior of the fluid between them can be written as
(2)$$\rho \frac{Dv_{\kappa }}{Dt}=\frac{\partial }{\partial x_{K}}\left[-p+(\lambda +\mu )\frac{\partial v_{\alpha }}{\partial x_{\alpha }}\right]+\mu \frac{\partial^{2}v_{\kappa }}{\partial x_{\alpha }\partial x_{\alpha }},$$
where *x* is the Cartesian coordinate and *v* denotes the velocity component.

By adding the continuity equation, Equation (2) can also be written as
(3)$$\rho \frac{Dv_{\kappa }}{Dt}=-\frac{\partial }{\partial x_{\kappa }}[p+(\lambda +\mu )\Delta ]+\mu \frac{\partial^{2}v_{k}}{\partial x_{\alpha }\partial x_{\alpha }}$$

Define the parameter *ε = h*_0_*/l*, and introduce a dimensionless coordinate system,
(4)$$X_{i}=\frac{x_{i}}{l},$$
(5)$$z=x_{3}/h_{0}=x_{3}/\varepsilon l,$$
where *x*_1_ and *x*_2_ denote the *x* coordinate and *y* coordinate, respectively, and *x_3_* denotes the *z* coordinate. The equations for the lateral and normal velocity components can be simplified, respectively, as follows:(6)$$\begin{array}{ll} \frac{\partial \pi }{\partial x_{i}} & =\partial^{2}u_{i}/\partial z^{2}-R_{s}P\left(\partial u_{i}/\partial T+W\partial u_{i}/\partial z\right)-\left(R_{s}+R_{L}\right)Pu_{i}\partial u_{i}/\partial X_{i}\\ & -\epsilon^{2}\left[\frac{\partial \theta_{s}/\partial x_{i}}{\left(1+V/\omega l\right)}+\frac{\partial \theta_{L}/\partial X_{i}}{\left(1+\omega l/V\right)}-\frac{\partial^{2}u_{i}}{\partial X_{i}\partial X_{i}}\right] \end{array}$$
and
(7)$$\begin{array}{ll} \frac{\partial \pi }{\partial z} & =\frac{\epsilon^{2}}{\left(1+V/\omega l\right)}[\partial^{2}w/\partial z^{2}-\partial \theta_{s}/\partial z-R_{S}P(\partial w/\partial T+w\partial w/\partial z)-\left(R_{S}+R_{L}\right)Pu_{i}\partial w/\partial X_{i}]\\ & -\frac{\epsilon^{2}}{\left(1+\omega l/V\right)}\frac{\partial \theta_{L}}{\partial z}+\frac{\epsilon^{4}}{\left(1+V/\omega l\right)}\frac{\partial^{2}w}{\partial X_{i}\partial X_{i}}, \end{array}$$
where *R_S_* and *R_L_* are the modified Reynolds number, *θ**_s_* and *θ**_L_* are the dimensionless dilatational stresses, *T* is a dimensionless time defined as *T* = *ωt*, *u_i_* is the velocity, *π* is the order unity of pressure, and *w* is the dimesionless velocities of order unity [18].

Theoretically, by solving Equations (6) and (7) mentioned above with initial conditions and boundary conditions, the fluid flow characteristics can be calculated completely. However, owing to the additional second-order item in the Navier–Stokes equation compared with other equations, it is difficult to obtain an exact numerical solution of fluid flow. In most cases of interest, the aspect ratio of a structure is large enough to ignore the terms of the second-order or higher in *ε*, so the right part of Equation (6) becomes zero and then the full equations become much simpler. With such a simplification, a single partial differential equation for the pressure in the isothermal gas film can be derived, which is called the Reynolds equation [24,25], listed as follows:(8)$$\frac{\partial }{\partial x}\left(\rho \frac{h^{3}}{\mu }\frac{\partial P}{\partial x}\right)+\frac{\partial }{\partial y}\left(\rho \frac{h^{3}}{\mu }\frac{\partial P}{\partial y}\right)=12\frac{\partial (h\rho )}{\partial t}.$$

By solving the Reynolds equation with the trivial pressure boundary condition under the assumption of a large aspect ratio, the damping coefficient for the rectangle plate and circle plate can be respectively derived as [12]
(9)$$c_{\mathrm{rec}}=\frac{\mu lw^{3}}{h^{3}}\beta (\eta )$$
and
(10)$$c_{\mathrm{cir}}=\frac{3\pi }{32h^{3}}\mu d^{4},$$
where $\beta (\eta )=16\eta \left(1-\frac{192\eta }{\pi^{5}}\sum_{n=\;\mathrm{odd}\;}\frac{1}{n^{5}}\tanh\frac{n\pi }{2\eta }\right)$ and *η* is the ratio *w/l*.

From the discussion above, it is obvious that the damping coefficient is derived with the neglect of second-order and higher items of *ε*, which means the film thickness is far smaller than the plate length. Hence, for most MEMS devices with large aspect ratio structures, the calculation result of the Reynolds equation is a rather good approximation of the Navier–Stokes equation and is well suited to describe the damping force acting on the devices. In other words, the original equation of Reynolds is the lowest order equation of the Navier–Stokes equation [25], thus the error of the Reynolds equation is also of the order of *(h/l)*^2^. If the length of the plate is not large enough compared with the film thickness, the damping force calculated by the Reynolds equation is underestimated, so that the finite size effect should be considered, which will cause the pressure variation across the film.

Figure 1 shows the static pressure of the air film central point along the z-axis in the simulation when the thickness of air film is equal to the length of the rectangle plate, which is 0.6 mm. The middle point of the air film is set as the origin of the z-axis. Obviously, the pressure distribution across the film is a parabola, and the pressure values at both ends of the parabola are not equal under the influence of the finite size effect, which contradicts the premise of the Reynolds equation.

## 3. Simulation of Squeeze Film Air Damping in Low Aspect Ratios

In this section, simulation models of squeeze film air damping are built to investigate the finite size effect of a low aspect ratio. Two basic structures in MEMS, including the rectangular structure and circular structure, are selected in this process. Besides, in order to distinguish the influence of the boundary effect and finite size effect on squeeze film air damping, four simulation models are introduced, which are the simple models with finite size effect only (the same as elongation model) and compact models with both finite size effect and border effect for rectangular plates and circular plates, respectively.

### 3.1. Simulation Model

First, the simple model with finite size only is built. The physical geometries of two shape structures in the simulation are shown in Figure 2. The main parameters set in the simulation are shown in Table 1. Taking the rectangular plate as an example, the upper face in the z-direction is set to a stationary wall, while the lower face serves as a moving plate with constant velocity. Besides, the four faces around are set to pressure-outlet regarding the trivial boundary, in which case the pressure is forced to zero in the boundary of the plate. The solid part inside represents the air between the wall and plate.

In order to investigate the finite size effect of squeeze film air damping quantitatively, the air film thickness is divided into three groups, 0.06 mm, 0.6 mm, and 6 mm, and the aspect ratio of the plates *γ* (*γ* = l/*h*), which is the reciprocal of *ε*, increases from 1:1 to 10:1.

The above simulation models consider the trivial boundary condition only, which means the finite size effect is considered only and the border effect is not taken into account. However, from the previous research on the border effect, it is obvious that the trivial boundary condition may cause an underestimation of the damping effect.

Thus, the compact simulation geometries (as shown in Figure 3) are also built, considering the border effect and finite size effect together to better understand their influence on squeeze film air damping simultaneously and utilize the damping model further. For example, taking the conditions of a rectangular plate, the outside part represents the fluid and the inside part represents the solid volume, like a rectangular plate. Similar to the simple model, the air film thickness of the two structures is divided into three groups, 0.06 μm, 0.6 μm, and 6 μm, and the aspect ratio of the plate increases from 1:1 to 6:1. The inner part is set to a moving plate with a magnitude of 10^−5^ m/s to simulate the linear motion. Besides, the upper face of the outside part is set to a wall, and the lower face and the other four faces of the outside part serve as pressure outlet symbols of the ambient pressure away from the moving plate, which means the non-trivial boundary condition. After initialization, the whole process was conducted by a steady simulation.

Regarding the simulation data analysis, we use the post-processing tool to obtain the damping force by integrating the pressure on the surfaces of the moving plate. As the damping force is the production of the damping coefficient and velocity, we can obtain the value of the damping coefficient by *c_d_ = F/v*.

### 3.2. Simulation Results and Modified Expressions for Squeeze Film Air Damping

#### 3.2.1. Finite Size Effect Only

The original theoretical damping coefficients for the rectangular plate and circular plate are derived from the Reynolds equation with trivial boundary conditions, and the expressions are listed as Equations (9) and (10). As a comparison, the simulation results and theoretical results of a rectangular plate and circular plate with the finite size effect only are shown in Figure 4. They are similar to the rectangular plate and circular plate in that the deviations between the simulation results and theoretical results are considerable when the aspect ratio of plates is not sufficiently large. When the thickness of the air film is equal to the length of the plate, the deviations between the original expression and simulation result reach 237.240% and 266.67% for the rectangular plate and circular plate, respectively, owing to the neglect of the finite size effect. Besides, the results demonstrate that the deviation between the theoretical results and the simulation results decreases dramatically with the increasing aspect ratio. When the aspect ratio of the plate is larger than 7, the relative error of the original expression is less than 5%.

Furthermore, Figure 5 depicts the static pressure of the air film central point along the z-axis with different aspect ratios for rectangular plates. It is likewise interesting to observe that the pressure across the film varies significantly in low aspect ratios. Besides, the pressure change also becomes smaller with the aspect ratio’s growth, which confirms that the finite size effect is related to the pressure change across the film.

It is obvious that Figure 4 shows that, as long as the aspect ratios for the rectangular and circular plate are fixed, the deviations of simulation results relative to original expressions almost remain the same, no matter the thickness of air film. It can be seen from the fitting curve that the deviation is proportional to the negative quadratic of *γ*, and the fitting coefficient for the rectangular plate and circular plate is 2.372 and 2.667, respectively. Thus, based on the original damping coefficient function and the fitting curve of deviation against the aspect ratio when the air film thickness is 0.6 mm, as shown in Figure 6, the squeeze film air damping coefficients for rectangular plate and circular plate with finite size effect only are expressed, respectively, as follows:(11)$$c_{\mathrm{rec}}=\frac{\beta \left(\frac{w}{l}\right)\times \mu \times l^{4}}{h^{3}}\times \left(1+2.372\gamma^{-2}\right)$$
and
(12)$$c_{\mathrm{cir}}=\frac{3\pi }{32h^{3}}\times \mu \times d^{4}\times \left(1+2.667\gamma^{-2}\right)$$

It can be seen from Figure 6 that the fittings of deviation are excellent with R^2^ = 1 for rectangular plates and circular plates. As a result, Figure 7 shows that the new functions reproduce the squeeze film air damping coefficients with a maximum relative error smaller than 1% compared with the simulation results, which shows the finite size effect is adequately included.

#### 3.2.2. Complete Model with Both Border Effect and Finite Size Effect

The simulation results discussed in Section 3.2.1 are based on the simple model with a trivial boundary condition, which means the border effect is not considered. Like in the previous research, the squeeze film air damping with non-trivial boundary condition can be replaced by a surface extension model with trivial boundary condition and the extracted elongation Δ*l* is almost constant (Δ*l =* 1.3*h*). That is to say, for a rectangular plate with width *l* and the air film thickness *h*, the damping coefficient with a border effect equals that for a rectangular plate with width *l +* 1.3*h* without a border effect. The known analytical expressions of squeeze film air damping for the rectangular plate and circular plate with a border effect are [17]
(13)$$c_{\mathrm{rec}}=\frac{\beta \left(\eta \right)\times \mu \times {\left(l+1.3h\right)}^{4}}{h^{3}}$$
and
(14)$$c_{\mathrm{cir}}=\frac{3\pi }{32h^{3}}\times \mu \times {\left(d+1.3h\right)}^{4}.$$

The simulation results for the compact model with both the border effect and finite size effect for the rectangular plate and circular plate are shown in Figure 8, respectively. It is demonstrated in Figure 8 that the original expressions for the border effect only cause a deviation of 52.998% and 67.473% between the simulation result and theoretical result for the rectangular plate and circular plate, respectively, when the aspect ratio is 1. Similarly, the error owing to the neglect of the finite size effect decreases with the increasing aspect ratio. It is interesting to observe that the deviations in the compact model are smaller than those in the simple model with the same aspect ratio. This can be explained by the elongation model of the border effect in that, because the squeeze film air damping with a border effect can be replaced by an elongation model without a border effect, as the aspect ratio becomes larger than the simple model, the relative error due to the finite size effect also becomes smaller. 

Besides, we also compare the simulation results of the compact model with the elongation model, although with the finite size effect included this time. The results shown in Figure 9 indicate that the elongation model’s deviations relative to the compact model’s results (which serve as the baseline of the true results) decrease dramatically by adding the finite size effect. By including the finite size effect into the elongation model, the whole error of the model drops from 40.289% to 10.189%.

Then, by subsitituting *l* with *l’ = l +* 1.3*h* and d with *d’ = d +* 1.3*h* in Equations (11) and (12), we can combine the elongation model with the finite size effect. Thus, we obtain the compact calculating formulas of squeeze film air damping for the rectangular plate and circular plate, respectively, as follows:(15)$$c_{\mathrm{rec}}=\frac{\beta \left(\frac{w}{l}\right)\times \mu \times {\left(l+1.3h\right)}^{4}}{h^{3}}\times \left(1+2.372{\left(\frac{l+1.3h}{h}\right)}^{-2}\right)$$
and
(16)$$c_{\mathrm{cir}}=\frac{3\pi }{32h^{3}}\times \mu \times {\left(d+1.3h\right)}^{4}\times \left(1+2.667{\left(\frac{d+1.3h}{h}\right)}^{-2}\right).$$

The error bar of new expressions is depicted in Figure 9 in the orange line, with a maximum of 7.700% for the rectangular plate and 10.185% for the circular plate. The new formulas for squeeze film air damping can be seen as compact theoretical solutions of the Navier–Stokes equation for rectangular and circular plates, which take the size and border effects into account.

## 4. Conclusions

The damping coefficient is a significant parameter in the design of MEMS devices, which motivates the need to modify the theory to improve the overall accuracy of the calculation. In this paper, we have a quantitative investigation of the finite size effect of squeeze film air damping through a series of simulations. Based on the simulation results of two different models with the original solution of the Reynolds equation, we obtain the quick-calculating formulas for squeeze film air damping with finite size only and including both the finite size effect and border effect, respectively. The lengths of structures used in our investigation vary from micrometer to millimeter, which shows the validity of new formulas in a scalable range. Besides, with the finite size effect taken into account, the deviations of the theoretical results compared with the simulation results are reduced by two orders of magnitude in the simple model with the finite size effect only and one order of magnitude in the compact model with the finite size effect and border effect, which greatly extends the validity of the squeeze film air damping model to the structures with lower aspect ratios. Compared with the previous model and design, our new formulas pave the way for a quick and accurate damping design for MEMS devices, especially those with lower aspect ratios, which is of great importance for the community.

## Figures and Tables

**Figure 1 sensors-22-01054-f001:**
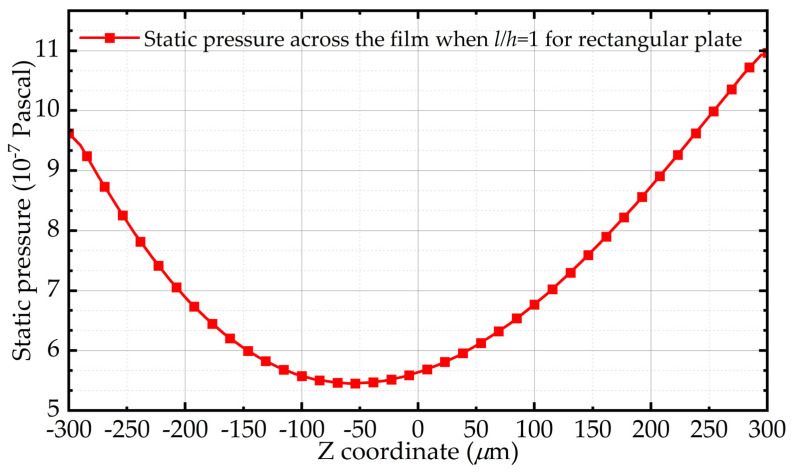
Distribution of static pressure along the thickness of air film in the simulation when the length of the plate and the film thickness are both 600 μm.

**Figure 2 sensors-22-01054-f002:**
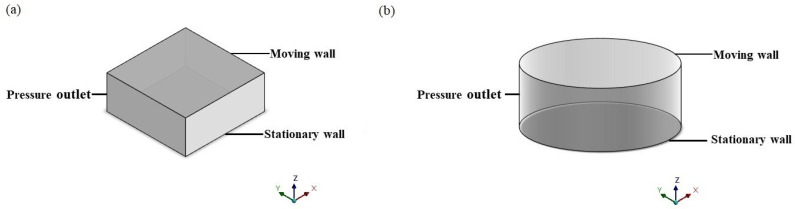
Physical simulation geometries of the (**a**) rectangular plate and (**b**) circle plate with the finite size effect only.

**Figure 3 sensors-22-01054-f003:**
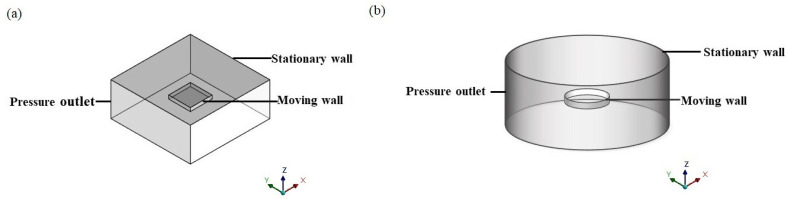
The compact simulation geometries for the (**a**) rectangular plate and (**b**) circular plate.

**Figure 4 sensors-22-01054-f004:**
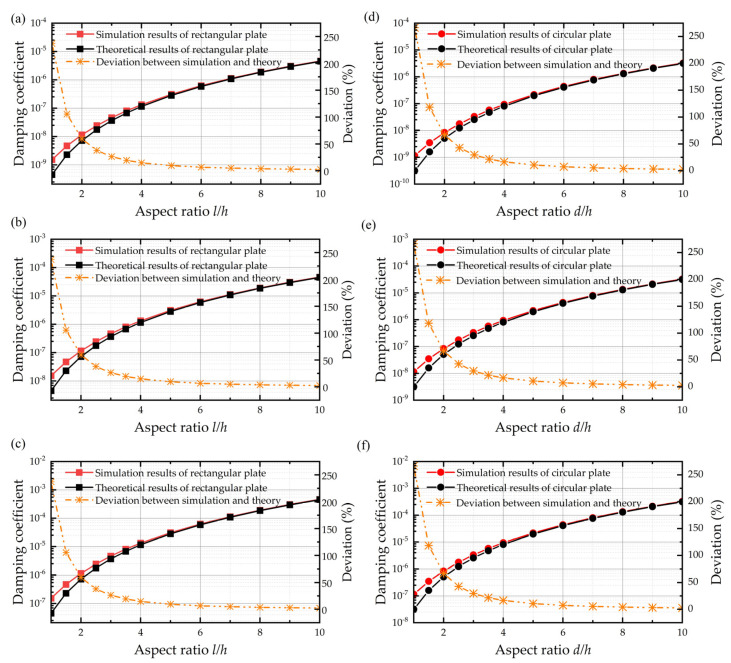
Damping coefficient of simulation results, theoretical results, and the deviations between them. (**a**–**c**) Rectangular plate when the air film thickness is 0.06 mm, 0.6 mm, and 6 mm from top to the bottom. (**d**–**f**) Circular plate when the air film thickness is 0.06 mm, 0.6 mm, and 6 mm from top to the bottom.

**Figure 5 sensors-22-01054-f005:**
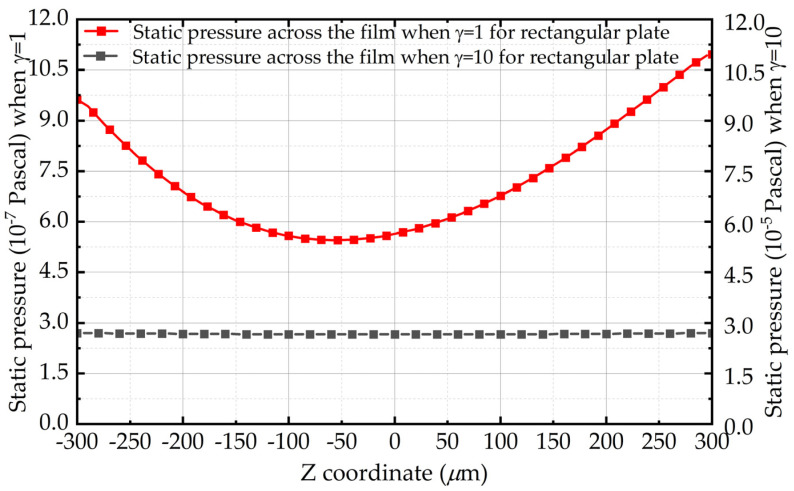
Pressure change across the fluid film with aspect ratio *l/h* for a rectangular plate when the air film thickness is 0.6 mm.

**Figure 6 sensors-22-01054-f006:**
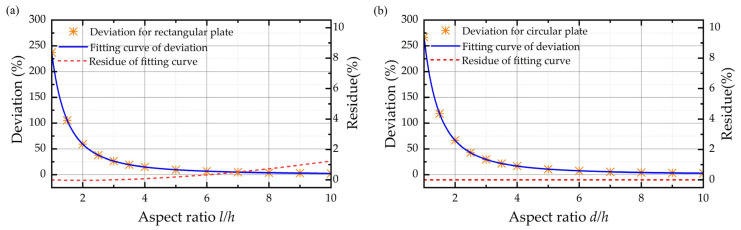
Fitting curve of deviations between the simulation results and theoretical results along with the residue of the fitting curve when the air film thickness is 0.6 mm for the (**a**) rectangular plate and (**b**) circular plate.

**Figure 7 sensors-22-01054-f007:**
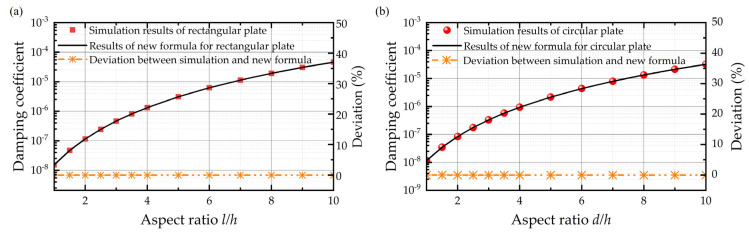
Damping coefficient of simulation results, fitting formulas, and the deviations between them for the (**a**) rectangular plate and (**b**) circular plate considering the finite size effect only when the air film thickness is 0.6 mm.

**Figure 8 sensors-22-01054-f008:**
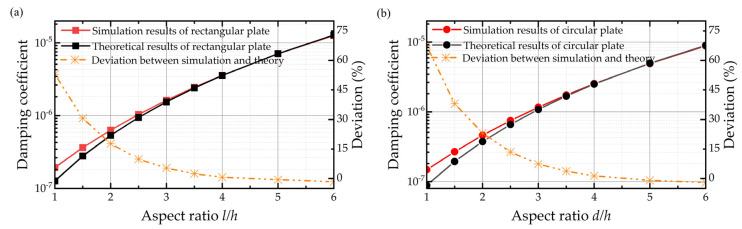
Damping coefficients of the simulation results and theoretical results and deviations between them for the (**a**) rectangular plate and (**b**) circular plate considering both the finite size effect and border effect when the air film thickness is 0.6 mm.

**Figure 9 sensors-22-01054-f009:**
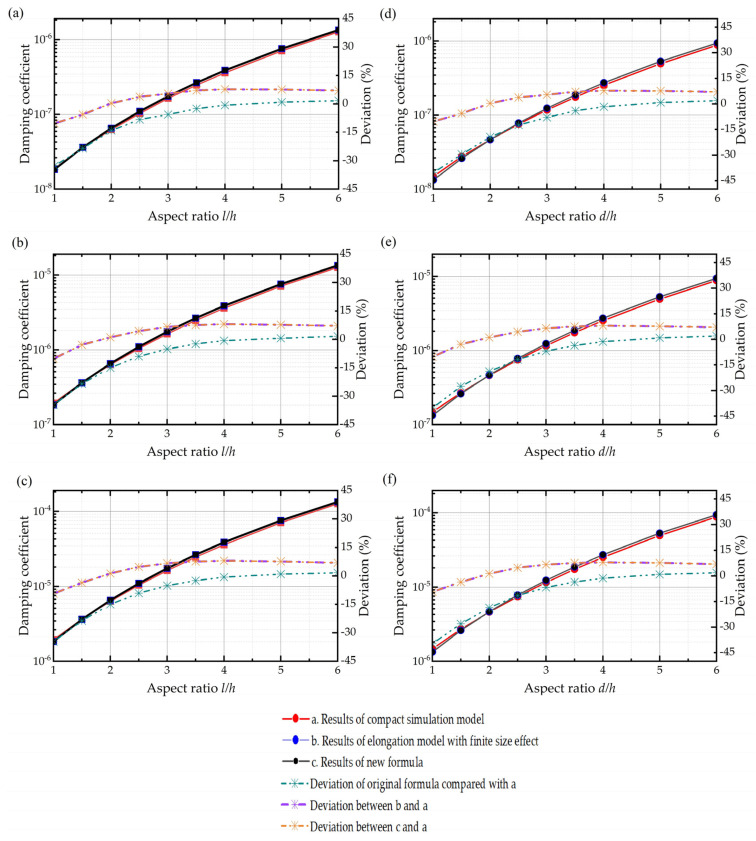
Damping coefficients of simulation results with the compact model and elongation model with the finite size effect and theoretical results of the compact modified calculating formula along with the deviations between them. (**a**–**c**) Rectangular plate when the air film thickness is 0.06 mm, 0.6 mm, and 6 mm from top to the bottom. (**d**–**f**) Circular plate when the air film thickness is 0.06 mm, 0.6 mm, and 6 mm from top to the bottom.

**Table 1 sensors-22-01054-t001:** Simulation parameters setting.

Parameter	Value
Ambient pressure, *P*_0_ (Pa)	1.01 × 10^−5^
Temperature, *T* (K)	300
Viscosity coefficient of air, *μ* (N × s)/m^2^	1.7894 × 10^−5^
Speed of the moving plate, *v* (m/s)	1 × 10^−5^

## Data Availability

Not applicable.

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
