# Peer review of "Extending the Validity of Squeeze Film Damping Models with Lower Aspect Ratios"

_sensors, 2022, doi:10.3390/s22031054_

Round 1

Reviewer 1 Report

Dear colleagues,

first many thanks for your contribution. The topic is interessting for development of dynamic MEMS and NEMS deices. You find below some issues with the text, which have to be improved.

Page 2.
Space [20] small errors
Page 3. What is B (4) in the equation?
Page 3. Equations 6 and 7. Authors have to provide more information for readers to understand it. Refences, derivation and variables in the equations.
Page 4. Figures with better quality and centered. Please provide information to the graph. It’s measurement data?
Page 5. Better figures 2 and 3. The pressure outlets are invisible in the figures.
Page 6. Better figure 4. Figure 5.
Page 7. Better quality figure 6.
Equations 11 and 12. More information and derivation from 9 and 10.
Page 8. Reference for equations 13 and 14.
Better figure 8.
Page 9. Better figures and explain the combination of two models to get the equations 15 and 16

Reviewer 2 Report

This paper presents a new squeeze film damping model to deal with the finite size effect and border effect, which is neglected in previous studies. From the comparison between the simulation and the theoretical results, the new formulas show high validity and practicality. This new model helps to instruct MEMS design. The manuscript may be accepted after minor modification. My detailed comments are as follows.

  1. In the title, the term ‘squeeze film damper’ is not proper and should be changed to ‘squeeze film damping. In MEMS devices, the energy dissipation needs to be avoided, while the damper is designed to absorb energy and suppress vibration.
  2. In Figures 4 and 9, there lack the subtitles of (c), (d), (e), (f). In Figure 6, the subtitles are also missing.
  3. What methods and software are used in the simulation, and how the simulation process is implemented?
  4. In Figure 6, please explain the reason for the difference in residual trends between rectangular and circular plates.
  5. The keywords cannot well summarize the focus of the paper.
